# Norwegian Community Pharmacists’ Experiences with COVID-19 Vaccination—A Qualitative Interview Study

**DOI:** 10.3390/pharmacy11060181

**Published:** 2023-11-19

**Authors:** Ragnhild Vold Aarnes, Marianne Kollerøs Nilsen

**Affiliations:** 1Department of Clinical and Molecular Medicine, Norwegian University of Science and Technology, 7491 Trondheim, Norway; ragnhild.v.aarnes@gmail.com; 2Boots Pharmacy Surnadal, 6650 Surnadal, Norway; 3Faculty of Nursing and Health Sciences, Nord University, 7800 Namsos, Norway

**Keywords:** COVID-19 vaccination, pharmacist, community pharmacy, Norway

## Abstract

Background: Immunising the population became important during the COVID-19 pandemic. Community pharmacies in Norway collaborated with municipalities to offer a vaccination services to increase the vaccination rate. Only some pharmacies were allowed to offer this service in the pandemic’s early phase. This study learns about pharmacists’ experiences during this first period of COVID-19 vaccination services in community pharmacies, which is relevant for informing policy and organisational decision makers about the feasibility and acceptability of pharmacy vaccination. Methods: Individual interviews were conducted with 13 pharmacists in community pharmacies offering a COVID-19 vaccination service. Informants were recruited from the eleven pharmacies that first offered COVID-19 vaccinations. The key themes in the interview were COVID-19 vaccination, what the pharmacists think about the vaccination service, and how it is performed. The data were analysed using systematic text condensation. Results: Three main themes and eight subthemes were identified. The main themes were creative solutions, organising and making resources available, and professionally satisfying and an important mission. The interviewed pharmacists experienced the COVID-19 vaccination service as hectic but something important that they would prioritise. They experienced their efforts to be substantial in the pandemic’s early phase. Conclusions: Pharmacists in community pharmacies were a resource for increasing the vaccination rate during the COVID-19 pandemic. The pharmacies’ easy accessibility and the pharmacists’ ability to adjust their daily workflow for a new service should be considered when an expanded healthcare service is needed.

## 1. Introduction

The worldwide outbreak of the contagious COVID-19 disease was declared a pandemic in March 2020. Several actions were implemented to limit the virus’s spread and prevent deaths. The international development of vaccines was a central action, and several countries began vaccinations in December 2020. In Norway, there was a national strategy for vaccination and who to prioritise, with municipalities being responsible for developing a system to implement this [1]. During this period, the speed of the vaccination process was reduced due to a lack of vaccinators, especially those able to immunise healthcare personnel. As more vaccines became available, the goal was to vaccinate as many individuals as possible to achieve population immunity.

In 2021, the Norwegian Association of Pharmacies signed an agreement with the Oslo municipality that piloted pharmacists in community pharmacies providing COVID-19 vaccinations after completing a training program to increase the municipality’s vaccination rate. The first group of patients were to be healthcare personnel [2]. After the initial period, more pharmacies received permission to offer vaccinations, and restrictions were relaxed on who could receive vaccination. Later, in December 2021, the government mobilised all Norwegian pharmacies to support COVID-19 vaccination.

Only physicians and nurses have traditionally been authorised to administer vaccines. However, some countries later allowed pharmacists to offer this service [3,4], even though turf wars with physicians exist [5]. Global variations exist in policies and regulations regarding pharmacists’ involvement in vaccination services, but the trend is to expand their role as vaccinators [3,6]. Studies have shown that including pharmacists in vaccination programs increases the vaccination rate, accessibility, and clinical outcomes [3,4,5,6]. Pharmacists in community pharmacies are trusted healthcare professionals who are easily accessible, since pharmacies are often located in neighbourhoods and have extended opening hours. Many individuals also prefer to be vaccinated in a pharmacy rather than in a general practitioner’s office [3,7]. In Norway, pharmacies have offered vaccination as a service since 2017, beginning with flu vaccines and later expanding to other vaccines. Pharmacies must ensure that the personnel administering vaccines complete a training program and are licensed as a vaccinator. A code of practice developed for vaccination in pharmacies must be followed. In Norway, pharmacists with Bachelor’s and Master’s degrees and pharmacy technicians can obtain a license as a vaccinator [8]. A special training program was developed for COVID-19 vaccination, with one entry criterion being that the vaccinators had to be previously certified for flu vaccination.

Notably, the first period of in-pharmacy COVID-19 vaccinations was while the country was still in lockdown. COVID-19 vaccination began as a pilot service. While some studies have examined pharmacists’ experiences with general vaccination [9,10,11,12], little is known about their experiences with COVID-19 vaccination during this initial period. Therefore, this study interviewed pharmacists participating in the pilot pharmacy COVID-19 vaccination program with the aim to learn about their experiences piloting a new service during this unique period. This could provide valuable insights into how and to what extent community pharmacies can change and adapt their practice quickly when needed, for example, in the event of a pandemic or epidemic. Experiences in this context are the pharmacists’ individual experiences and perceptions about the matter. The following research question guided this study: what were the early experiences of pharmacists in Norwegian community pharmacies with COVID-19 vaccination?

## 2. Materials and Methods

This study used a descriptive qualitative design to obtain a straight description of the phenomenon of pharmacist-led COVID-19 vaccination [13], with individual semi-structural interviews, which were well suited to restructure the pharmacists’ perceptions and experiences [14]. It also used the COREQ checklist (Appendix A) [15] to improve comprehensive and explicit reporting. The first author conducted the interviews, prepared the interview guide, and conducted the analysis in regular discussion with the second author.

### 2.1. Study Setting, Sampling, and Participants

An individual responsible for COVID-19 vaccination in the Norwegian Association of Pharmacies was contacted for information about which pharmacies offered this service. Informants were recruited from pharmacies in Oslo, Norway, where the pharmacy COVID-19 vaccination program was piloted. The study used a strategic sampling strategy [16], since the informants should have experience working in a pharmacy offering COVID-19 vaccination. Its inclusion criteria were: pharmacists working in a pharmacy that offered COVID-19 vaccination in the early phase of the pandemic. 

From the individual responsible, we received a list of eleven pharmacies. All eleven pharmacies were contacted by e-mail in September 2021, introducing this study with an attached information letter encouraging pharmacists to volunteer to participate. The pharmacies that had not responded were reminded via telephone in November 2021. The head of each pharmacy, probably the person receiving the e-mail, gave us the names and contact information of the pharmacists in their pharmacy who would like to participate in this study. We successfully recruited 13 pharmacists from nine pharmacies. 

Saturation [17] was used to guide the number of interviews; the sample size was sufficiently large for elucidating the aims of the study, even if it could be more varied, and the researchers did not find significantly new information when analysing the final two interviews. In autumn 2021, only a few pharmacists had experience with COVID-19 vaccination in Norway. Participation was voluntary. Written informed consent was obtained from each informant before the interviews started. 

### 2.2. Data Collection

The first author developed a semi-structured interview guide inspired by research on vaccination in pharmacies [17,18] and themes relevant for the research question. The interview began by informing the interviewee about the research study and asking for background information before asking general questions about pharmacists and vaccination. The main part of the interview was about COVID-19 vaccination, encouraging the pharmacists to share their thoughts about participating in the vaccination program during the pandemic and express their experiences regarding vaccination. Some of the questions in the interview guide were: What do you think about pharmacists offering vaccine services (both in general and for the COVID-19 pandemic)? Could you tell us about your perceptions and experiences of contributing in a situation of crisis, such as in the ongoing pandemic? Could you tell us about how you perform COVID-19 vaccination, and the training program? How do you think the new service affects the daily work? Follow-up questions encouraged participants to clarify or elaborate on their responses. The interview guide was validated through a pilot interview; this interview was not included in the analysis. The pilot interview was to test the interview guide, if the questions were relevant and understandable. Only minor language revisions were made afterwards. The initial analyses for the first interviews provided a new understanding, and the order of questions was revised for the subsequent interviews. However, the main questions were the same in all interviews. None of the interviews were repeated.

The interviews were performed face-to-face between October 2021 and January 2022 and lasted between 15 and 45 min. Only the informant and the first author participated in the interviews, which took place in the pharmacy. The interviews were audio-recorded and transcribed verbatim, and were deidentified and anonymised once transcribed. The audio recordings were encrypted while stored and deleted after completing the analysis. No further member check was conducted.

### 2.3. Data Analysis

The analysis was based on systematic text condensation [16]. This method is well-suited for a descriptive study. It is a four-step process, seeking similarities and differences in the data material. In the first step, preliminary themes were labelled in the transcripts. In the second step, the transcripts were coded according to these themes by identifying units of meaning, and the main themes were adjusted. In the third step, the units of meaning were grouped into subthemes, and a condensate was made of each subtheme. In the fourth and final step, an analytic text was synthesised based on the condensates of each theme and subtheme. Both authors frequently discussed the steps to ensure the interpretation of our findings. 

### 2.4. Ethical Considerations

Due to the interview recordings, a notification form for personal data regarding the study was sent to Sikt (The Norwegian Centre for Research Data; Ref: 349157). The study adhered to the ethical principles for research based on the Declaration of Helsinki [19]; participation was voluntary, the informants were competent to consent, and written informed consent was received from all informants. The informants could withdraw at any time during the interview; no one did.

## 3. Results

Table 1 describes the study sample:

The analysis identified three main themes describing the pharmacists’ experiences with COVID-19 vaccination: creative solutions, organising and making resources available, and professionally satisfying and an important mission, as shown in Table 2

### 3.1. Creative Solutions

The pharmacists described the practical challenges they had to overcome, often within a short time. Creative solutions were especially required regarding the short expiration of the vaccine doses and individuals remaining for post-vaccination observation, which was challenging in small pharmacies given social distancing requirements.

The pharmacists did not want to waste any vaccine doses. However, there were strict regulations about prioritising who should receive the doses at that time. They described experiencing conflict between doing what the government had decided and deciding what to do about leftover doses. Some informants described how they set a goal never to throw away doses, resulting in leftover doses being used for the pharmacy employees or generally random individuals visiting the pharmacy. Some also described how, when people did not attend their appointment, pharmacy staff searched the nearby area to find someone who could have the vaccine. Most pharmacists expressed that always having to think about finding a solution to avoid wasting doses was chaotic and demanding. The following quotation illustrates this situation:


*‘Then we had one dose left, and I didn’t know what to do with it. Then I ran over to the grocery store next door and shouted “Hi, is there anyone who needs vaccine?” The cashier needed one, and she went with me and got it’.*
(Pharmacist 12)

Another challenge the pharmacists experienced as demanding was the requirement for individuals to wait 20 min after vaccination, given the two metre social distancing requirement due to the COVID-19 pandemic. The pharmacies were often small and not adapted to these situations. One pharmacist stated that pharmacies should not be used this way; vaccinators should instead go to large vaccination facilities in the municipalities. Pharmacists in some larger pharmacies did not experience these requirements as challenging. One pharmacist working in a smaller pharmacy described how they solved it:


*‘In the beginning, it was really difficult. Two-meter distance is difficult to achieve. What we did was to open the pharmacy before opening hours for vaccination and move the chairs to the correct distance using more than the vaccination room in the pharmacy. When the requirement became one meter, we could do the vaccination during opening hours, but just a few vaccinations at a time as the distance requirement was really important to uphold’.*
(Pharmacist 12)

### 3.2. Organising and Making Resources Available

Experiences of organisation and how a new vaccine service affected the daily work engaged the pharmacists. Having sufficient individuals at work and how the workflow was organised were important. Good systems were important to plan the daily workflow. Offering COVID-19 vaccination on top of other tasks affected pharmacy routines. The pharmacists stated that it was important for them that COVID-19 vaccination was prioritised.

Several pharmacists experienced that COVID-19 vaccination was more time demanding than flu vaccination, at least initially. Being able to offer the COVID-19 vaccination services affected the routines of pharmacy staff, especially who should do what at which time. They had to set new routines for the vaccination process to run smoothly. Some pharmacies had a sort of COVID-19 vaccination section. Several informants experienced the importance of having sufficient personnel at work to achieve a good workflow. Despite the lockdown and pandemic regulations, they experienced more people attending pharmacies during this period. As one informant stated:


*‘If we have personnel enough, one vaccinator could carry out the service, and it works well. When we are short of personnel, and everybody has to do everything, you have to be here and there and everywhere at the same time, the service will be messy’.*
(Pharmacist 3)

To organise the practice around COVID-19 vaccination, several pharmacists highlighted the ‘Helseboka’ system (Book of Health). This system made it possible to plan the vaccinations during the day. It was used to book appointments and help the pharmacists to know when patients came and who they were. Unlike flu vaccination, COVID-19 vaccination was not a drop-in service during this period. Using this system, they could plan vaccinations according to the staffing schedule and prepare the vaccine doses to prevent unnecessary waste. One informant described it like this:


*‘Unlike other vaccination that often are drop in, we could plan. And then we would know how to use our resources at all times. If we were clever enough; we should be able to be that, and we had the resources (staff) available and sufficient time, there was no problem’.*
(Pharmacist 5)

Most pharmacists considered the COVID-19 vaccination period as one where pharmacy tasks shifted and were not as challenging. They did not think the COVID-19 vaccination service negatively affected other tasks, even if they were sometimes busy. Maybe some customers had to wait slightly longer, and some administrative work was postponed, but it was not experienced as negative. The informants expressed that they wanted to prioritise COVID-19 vaccination and made room for it. One informant stated:


*‘It seems like there was a gap, which it (the COVID-19 vaccination) filled. It seems like it was room for it; we just did not realise it beforehand’.*
(Pharmacist 8)

### 3.3. Professionally Satisfying and an Important Mission

The pharmacists found participating in COVID-19 vaccination during the pandemic satisfying. The new task made them feel more important and contributing to society in a challenging time. This new task also broke up their daily work routines, allowing them to do and learn something new. Finally, several informants described how this made the pharmacist profession more visible in society.

Several informants expressed feeling pleased about providing COVID-19 vaccination; they felt more important than before. They became a resource for society, not just a drugstore. Some pharmacists also described how they contributed to vaccinating the population faster, and because the wait for the vaccine was long, the task became extra important. Two quotations describe this:


*‘It was good to feel that I actually could do something. Normally you just sit down and do nothing, keeping away from people.’ (Pharmacist 1); ‘I felt the joy of contributing in something so large, in the critical situation. I was really a proud pharmacist then’.*
(Pharmacist 8)

Since this was the pilot and first time using pharmacists during the pandemic, guidelines did not exist. Some pharmacists participated in all the steps of preparing the code of practice for COVID-19 vaccination, which The Norwegian Pharmacy Association prepared while engaging pharmacies through digital meetings. As part of a new and important service, several informants found this change in daily routine satisfying; it gave them new motivation to work in a pharmacy. The pharmacists hoped that COVID-19 vaccination in pharmacies was a service they could also offer after the pilot period. As one stated: ‘*The last day was almost wistful*’. (Pharmacist 1)

Several informants mentioned how this service had affected the visibility of the pharmacist’s role. They described how this altered the impression of pharmacists and was an eye-opener for the government to include pharmacists more in healthcare systems. One pharmacist described how the general population became more aware of the pharmacies’ services and that vaccination in pharmacies was more readily available than going through general practitioners. Several informants experienced getting positive feedback from both patients and general practitioners. Some informants described general practitioners who initially were sceptical but now did not hesitate to send their patients to pharmacies for vaccination. The following quotation illustrates how several pharmacists experienced a change in views towards them:


*‘I think many people see us like staff in a store, more than healthcare professionals. So, when we do this (the COVID-19 vaccination) and the people realise that these people know something about health. And, of course, we get all sorts of questions when the patient sits in the room and get their vaccine. So I think this service is a good advertisement for us pharmacists’.*
(Pharmacist 4)

## 4. Discussion

This study showed that pharmacists’ searched for creative solutions to use all vaccine doses and fulfil the set requirements. In addition, they adjusted their workflows and made resources available for this new service. The pharmacists considered contributing to COVID-19 vaccination satisfying and important for themselves, the profession, and society.

The two vaccines used in pharmacies, Spikevax^®^ (Moderna, Madrid, Spain) and Comirnaty^®^ (BioNTech-Pfizer, Mainz, Germany), had limited stability after dilution and vial puncture; if the doses were not used within a limited time, they had to be discarded as waste. This study found that there was almost a race against time to avoid this, and using all doses was highly prioritised. Another study in the United States also mentioned avoiding missing doses [20]. The focus on exploiting the vaccines may also be seen in their distribution at that time. The focus in society was to vaccinate as many as possible in the shortest amount of time, and avoiding wastage was important [1]. Wastage is not an issue for most flu vaccines, since they are delivered as prefilled syringes; in time, COVID-19 vaccines may also come in prefilled syringes, negating this challenge. When this study was conducted, the supply of vaccines was still somewhat unstable, and deliveries were smaller in some months [21], potentially contributing to the pharmacists’ focus on thinking of creative solutions. At the same time, these creative solutions entailed internal conflicts within some of the pharmacists. However, whether some creative solutions were ethically correct practices should be discussed.

Government regulations due to the pandemic affected pharmacy practice. How to adjust distances between individuals and the time patients were required to wait after vaccination also challenged the pharmacists to find new solutions depending on the size of their pharmacy. The fact that pharmacists offered vaccination before opening hours shows a high willingness to offer this service, and it was considered as important to provide a good service. A quantitative study among pharmacists in Saudi Arabia also showed a high willingness to vaccinate individuals against COVID-19 [22].

A new service will affect the daily workflow. While this change seems challenging, the pharmacies could still offer all necessary services to their patients and customers. Unlike in other countries where only pharmacists can give vaccinations, technicians in Norwegian pharmacies can also vaccinate individuals after completing a teaching program [20]. Like other studies [9,20], this study also found a lack of personnel as a barrier to vaccination in pharmacies. Good systems for workflow were seen as a facilitator. The pharmacists interviewed in this study highlighted the program they had for booking before potentially discussing the personnel situation. All the interviewed pharmacists considered the COVID-19 vaccination to be a new task that did not interfere with their other tasks. They also wanted to prioritise COVID-19 vaccination. The pharmacists seemed to find flexibility within their workflow when they believed the task was important. It must be considered that individuals were supposed to stay home during this period, and even when the pharmacies were allowed to open, they might have had fewer customers.

The final main theme was how the pharmacists found COVID-19 vaccination satisfying and important. This attitude also influenced the other two main themes. Even when the pharmacists saw barriers and difficulties, the advantages overshadowed them, consistent with other studies [20,22,23]. Another study exploring pharmacists administrating flu vaccines also highlighted the expanded scope of practice and professional satisfaction as key findings [24]. The pharmacists interviewed in this study experienced professional satisfaction with COVID-19 vaccination. They expressed a need for more professional challenges. From their statements, it is reasonable to believe that they would invest substantial efforts to make pharmacies a natural part of the national vaccine program, also for COVID-19 vaccination.

From these results, pharmacists seem to be searching for new roles, where COVID-19 vaccination showed society that community pharmacists are important in a pandemic and can quickly adapt to new roles. A scoping review of novel roles adopted by community pharmacists during the COVID-19 pandemic stated that it led to a paradigm shift in community pharmacies, where pharmacists expanded their roles and became key stakeholders in the healthcare system [25]. In particular, the pandemic sped up the shift in roles, and community pharmacies have shown that they can contribute to immunisation as a supplement to the rest of the healthcare system [25]. This is in line with the results in this study. A review supported pharmacist vaccination services since they are accepted by pharmacists and patients, increase vaccine rates, and improve vaccine access for the population [26]. A study in France found that pharmacists in community pharmacies contributed significantly and were important during the COVID-19 pandemic [23]. Community pharmacies may play a key role in vaccination programs, since they provided good experiences and effectively delivered essential and advanced services during the COVID-19 pandemic [27,28].

Pharmacists in other counties also found COVID-19 vaccination meaningful [29]. In our results, several pharmacists described receiving positive feedback. Other studies also found satisfaction among patients receiving the COVID-19 vaccine in pharmacies [7,29,30]. There are sometimes disputes among healthcare personnel on who can and cannot provide vaccinations [31]. In one study, general practitioners expressed relief in their workload when pharmacies offered COVID-19 vaccination and were satisfied with pharmacists’ role in COVID-19 vaccination [29]. Notably, some general practitioners changed their attitudes towards the vaccine service offered by community pharmacies. To be seen and make a difference are two fundamental needs for individuals at work, since they can be a source of feelings of mastery and happiness at work, positively affecting general health [32]. This study indicated that pharmacists desire to be a professional group to count on in the healthcare system. They contributed to society during a demanding period due to the COVID-19 pandemic, which they considered to be substantial. These findings are something the pharmacist profession itself should consider when discussing what should be pharmacists’ role.

### Strengths and Limitations

This study aimed to learn about the experiences of pharmacists testing a new service at a unique time. While its data were collected from a small group of pharmacists, the pilot COVID-19 vaccination program was conducted only in Oslo, and this group represents several of its pharmacies. We successfully included informants with a diversity in gender, education, and experience. However, whether these results are transferable to all other municipalities under non-pandemic conditions is unknown. Another limitation may be that only one informant was not licensed for vaccination, even though their pharmacy offered this service. Therefore, the answers from this group of pharmacists are inadequate.

As mentioned in the Methods, an individual from the Norwegian Association of Pharmacies assisted us in contacting pharmacies. We do not know how they selected pharmacies, and we do not know whether the collaborating pharmacies have a good or bad relationship with the Association or if some pharmacies were omitted for other reasons.

One possible limitation of this study is the validity of the sampling and the interview. Some of the pharmacists worked in the same pharmacy, which might have influenced what they talked about and how they described their vaccination practices. The data could be biased by this factor. Further validation with the interview guide and analysis with member checking would be preferable.

The researchers were not associated with the participating pharmacies. The first author is a Master’s student with a previous Bachelor’s degree in Pharmacy, and the second is an Associate Professor (PhD) at Nord university; both are female. The first author has some experience working in a pharmacy during the pandemic but not as a vaccinator, which could be both a strength and a limitation. Lack of personal experience with the vaccine service may limit the authors’ understanding of this practice, which might be a strength, since contextual factors will not bias their interpretations.

The informants talked willingly during the interviews. The informant–researcher relationship did not seem affected by the interviewer being a Master’s student. The quality of the interviews and analysis depends on the researcher’s experience [33]. Since the first author had limited research experience, this may have affected some of the initial interviews and analyses. Both authors discussed all steps in the research processes for validation, where the second author is more experienced with research.

## 5. Conclusions

Pharmacists’ contributions during this early phase of the COVID-19 pandemic indicate that pharmacists in community pharmacies are a resource for the healthcare service that was instrumental in increasing the COVID-19 vaccination rate. The pharmacists interviewed in this study experienced COVID-19 vaccination to influence their daily work, but their willingness to offer this service and contribute to vaccination during the pandemic was substantial. They demonstrated the ability to rapidly shift work with creative solutions and reorganisation to release the required resources. These results also show that this new role was professionally satisfying for pharmacists and important for society as the vaccination rate increased. The pharmacists experienced being important resources during the COVID-19 pandemic, and they were happy to feel that their effort made a difference.

Further research is needed to elaborate the extent to which pharmacists can contribute to the immunisation of the population, not just in a pandemic, but also regarding other vaccines.

Based on pharmacists’ willingness to adjust their service according to the needs of a new service, the government should consider making greater use of pharmacists and community pharmacies as a resource when there is a shortage of other healthcare personnel, as in this case of increasing vaccination rate in a pandemic. We find this study to be a contribution to the discussion of pharmacists’ role as immunisers and contributors to COVID-19 vaccination rates, to allow for a sustainable pharmacy vaccination service in Norway. It also gives new knowledge about expanding pharmacists’ role where we might see a paradigm shift coming up.

## Figures and Tables

**Table 1 pharmacy-11-00181-t001:** Description of study participants.

Gender	Age (Years)	Work Experience as a Pharmacist (Years)	Education	Certified as a Vaccinator
Male: *n* = 3 Female: *n* = 10	20–34: *n* = 3 35–49: *n* = 7 50–64: *n* = 3	1–10: *n* = 4 11–20: *n* = 4 21–30: *n* = 4 31–40: *n* = 1	Bachelor’s degree: *n* = 5 Master’s degree: *n* = 8	Yes: *n* = 12 No: *n* = 1

**Table 2 pharmacy-11-00181-t002:** Overview of the main themes and subthemes.

Main Theme	Creative Solutions	Organising and Making Resources Available	Professionally Satisfying and an Important Mission
Subtheme	Unused vaccine doses	Workflow	Satisfying to contribute
	Requirements	System for organisation	Professionally evolving
		Shift in tasks	Being visible

## Data Availability

The datasets from the current study are not publicly available due to the lack of consent from the informants for publicly sharing, transcripts (in Norwegian language) are available from the corresponding author on reasonable request.

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
