# Peer review of "Norwegian Community Pharmacists’ Experiences with COVID-19 Vaccination—A Qualitative Interview Study"

_pharmacy, 2023, doi:10.3390/pharmacy11060181_

Round 1

Reviewer 1 Report

Comments and Suggestions for Authors

Overall: Despite good intensions, I think this study needs several improvements before being publishable. Firstly, I don’t see any justification of it. What is the problem that it is trying to solve? Secondly, the results describe very practical and often obvious issues and I am not convinced where are the new relevant take home messages for an international audience? The underpinning of the study is weakly described. In the introduction, there is no reference to state of the art, and how the study builds on this knowledge. Thus, it is unclear what is the specific aim of the study (and the conducted interviews?!). Several parts of the methods are further insufficiently described.

Specific comments:

Abstract

Background: add ’In Norway’ + elaborate why it is relevant to learn about ‘…pharmacists’ experiences during this first period of the Covid-19 vaccination service in community pharmacies’.

Methods: missing key themes in the interview + how were participants recruited?

Introduction:

I don’t see any argument of why this study is needed? What is the problem that needs to be fixed?/ What is the added value that this study can bring? Why is the early phase in particularly important – and how does it differ from other phases? Hence, the sentence ‘While some studies have examined pharmacists’ experiences in general vaccination, little is known about their experiences with Covid-19 vaccination during the initial period’ – needs to be expanded including providing state of the art of existing knowledge in that specified area. Experiences as a a concept is too broad and has to be specified.

And why is vaccination so favorable – aren’t here any downsides to it, for example does it truly support the need for transformation of pharmacies into a place in the health care system where the individual medical health matters of patients can be discussed in detail?

Method

Please specify which perceptions and experiences around covid vaccination that were relevant for this study (according to a revised aim) – and then justify the choice of method accordingly. Please specify ‘descriptive qualitative design’ and why you aimed for that type of study?

More information is needed about how you selected participants (11 pharmacies out of how many – and why those 11?) – I assume many more than those participated were eligible for participation? Only when reading the limit section does the reader realize that it is the Norwegian Association of Pharmacies who selects and contacts the pharmacies – that must be stated upfront. It is also unclear until the limit section who is ‘the head of the pharmacies’! Further, it should be explicated how that person motivated pharmacists to participate? Was ‘experience’ the only selection criteria and why did you include several pharmacist from the same pharmacy – wouldn’t that scew data because when more participants work in the same pharmacy, then some results for example having to little room (‘individuals remaining for post-vaccination observation’) would be overrepresented in the data? This would not be a problem with an in-depth analysis but as this is a descriptive and broad analysis, then it is more problematic.

Exactly what type of ‘saturation’ was applied and how did you assess that this particularly form of saturation was indeed achieved?

Table 1 should be moved to the first part of the Result section.

The description of the core of the interview guide is insufficient: ‘Covid-19 vaccination, encouraging the pharmacists to share their thoughts about participating in the vaccination program during the pandemic and express their experiences regarding vaccination.’ As part of this expansion, please specify which literature was included in the interview guide as well as why and how.

What exactly was learned from the pilot interview – and which other learnings were done from analysis of the ‘first interviews’ to inform the later? And does that align with ‘the frame of the interview guide was the same in all interviews’?

I would reconsider which type of method you used. In the beginning of the method section it reads ‘individual in-depth interviews’ – but the interviews lasted between 15!!-45 minutes – so I don’t believe they can be called in-depth.

Results:

The process of analyses seems to have be carried out satisfactory – hence the results draw on trends of experiences across participants.

However, even though the design of the study is descriptive, the results for a large part deals with practical experiences and to a less degree with participants’ perceptions around it. Results like ‘Good systems were important to plan the daily workflow’ or ‘They had to set new routines for the vaccination process to run smoothly ’ or ‘Several informants experienced the importance of having sufficient personnel at work to achieve a good workflow’ - are quite obvious and hardly a surprise for anyone. In stead, I think the study should elaborate on the perceptions for example in regard to: ‘They described experiencing conflict between doing what the government had decided and deciding what to do about leftover doses.’ (p.5, l.154-155) or ‘Several informants expressed feeling pleased for providing the Covid-19 vaccination; they felt more important than before. They became a resource for society, not just a drugstore’ (p.6, l.226-227). The authors could also go more into depth with a topic mentioned only in the discussion ‘A scoping review of novel roles adopted by community  pharmacists during the Covid-19 pandemic stated that it led to a paradigm shift in community pharmacies, where pharmacists are key stakeholders in the healthcare system ‘ – hence what is involved in these paradigm shift – what did you find in the data?

Were there differences in opinions between bachelors versus masters’ degrees?

Discussion:

I wonder how relevant is the situation of ‘creative solutions to use all vaccine doses’ – what learnings can be derived from the results that can be used going ahead? Is it a situation specific to covid-19 or does it bear more general learning outcomes? The authors even themselves write in the discussion that ‘Wastage is not an issue for most flu vaccines since they are delivered as prefilled syringes; in time’ – hence is it a relevant theme to engage in – in the result section and discussion?

I don’t understand the sentence of ‘This study showed that pharmacists desire to be more important as a professional group’ and I am not convinced that it can be claimed based on the presented data.

Comments on the Quality of English Language

A bit a further proof reading would be good.

Reviewer 2 Report

Comments and Suggestions for Authors

The manuscript as submitted describes the challenges and most importantly the solutions that faced pharmacists in Norway related to mandates to institute vaccination. Its of interest to read the comments from the subjects in the project as they reflect many of the concerns faced by other medical enterprise entities. Of note involvement of our Pharmacists at least in the United States has been robust and significant as vaccinations can be readily obtained in virtually all commercial pharmacies. I compliment the Norwegian pharmacists on there approach to solutions provide vaccinations to their population. I do have some concerns as follows:

1. Was the study reviewed by a Norwegian Human Studies committee- IRB - in which issues related to protection of the subjects was addressed? 

2. The study design is a 'Qualitative descriptive design" with the PI developing the interview guide but also conducted the interviews with the pharmacists?  This is concerning for investigator bias- authors need to address how this was evaluated. Usually a separate trained group would conduct the interviews and collect the responses. As this is qualitative data it is important that no collection bias be present

How did the authors protect from cross contamination among their pharmacists? Interviewing a pharmacist may result in that pharmacist sharing their interview experience with their colleagues that would be also interviewed. 

2. It is also standard in qualitative studies to provide  "triangulation" to the use of data sources to support validity. Can the author provide additional insight as to validation for their process. 3. It appears the interview guide was validated only with a "pilot interview" with no experimental details. No focus group interviews? Please provide details on this pilot interview process and how this supported validation. 

The discussion is insightful and relates to a potential reader of this article how these very resourceful pharmacists sought solutions to get the job done. Rather than falling back of excuses for not achieving results the pharmacists sought and achieved solutions and as such achieved results and achieved personal and professional satisfaction in the process. As the role of the clinical pharmacist continues to evolve and accept more primary responsibilty for health care delivery - the study clearly illuminates how these professionals can be better integrated into the healthcare delivery workflow in Norway  

Comments on the Quality of English Language

Title should be rewritten to be more " rigorous" also some minor editing required

Round 2

Reviewer 1 Report

Comments and Suggestions for Authors

Requests by reviewer has been adequately dealt with

Author Response

Thank you for revising my article. Regards, Marianne K Nilsen

Reviewer 2 Report

Comments and Suggestions for Authors

I do accept the revisions and suggest the article be accepted in current revised form

Comments on the Quality of English Language

I do accept the revisions and suggest the article be accepted in current revised form

Author Response

(The authors gave the same response as above.)
